# Predictive Factors of the Fatality of Motor Vehicle Passengers Involved in Far-Side Lateral Collisions: A National Crash Database Study

**DOI:** 10.3390/healthcare11101496

**Published:** 2023-05-21

**Authors:** Akane Masumitsu, Masahito Hitosugi, Mineko Baba, Mami Nakamura, Kaoru Koike, Hitoshi Ida, Masashi Aoki

**Affiliations:** 1Department of Critical Care Medicine, Kyoto Medical Center, 1-1 Mukaihata-cho, Fukakusa, Fushimi-ku, Kyoto 612-8555, Japan; akanemas@belle.shiga-med.ac.jp (A.M.); kkoike@kuhp.kyoto-u.ac.jp (K.K.); 2Department of Legal Medicine, Shiga University of Medical Science, Tsukinowa, Seta, Otsu 520-2192, Japan; mamin@belle.shiga-med.ac.jp; 3Center for Integrated Medical Research, Keio University School of Medicine, 35 Shinanomachi, Tokyo 160-8582, Japan; mineko@keio.jp; 4Toyoda Gosei Co., Ltd., 1 Haruhinagahata, Kiyosu 452-8564, Japan; hitoshi.ida@toyoda-gosei.co.jp (H.I.); masashi.aoki@toyoda-gosei.co.jp (M.A.)

**Keywords:** motor vehicle passenger, lateral collision, fatality, NASS/CDS, injury severity

## Abstract

Although the risks faced by passengers in near-side lateral collisions are understood, and despite the presence of side airbags for injury prevention, passengers involved in far-side lateral collisions also suffer serious and fatal injuries. The objective of this study was to determine the independent predictive factors of fatality of motor vehicle passengers involved in far-side lateral collisions. Using 2010 records from the National Automotive Sampling System/Crashworthiness Data System (NASS/CDS), we selected 86 fatal and 325 non-fatal passengers with an Abbreviated Injury Scale (AIS) score of 2 or more. The background and injury severity of the passengers and collision characteristics were compared between the two groups. In a multivariable logistic regression analysis, variables independently associated with fatalities were female sex (Ref, male) (odds ratio [OR], 0.396), age (OR, 1.029), body mass index (OR, 1.057), total delta-V (OR, 1.031), head AIS score (OR, 1.679), chest AIS score (OR, 1.330), and abdomen AIS score (OR, 1.294). This is the first report to determine factors affecting fatality in passengers involved in far-side lateral collisions. Improving the safety of the vehicle interior, such as by including additional seatbelt systems or a side airbag that deploys between seats, might help to avoid fatalities, and reduce injury severity.

## 1. Introduction

A vehicle collision involving a lateral impact to a vehicle is one of several modes of crash and often has serious consequences for the passengers. A report from the United States suggested that lateral collisions resulted in 337 fatalities per 100,000 collisions and are thus more lethal than front collisions (321 fatalities per 100,000 collisions) and rear collisions (71 fatalities per 100,000 collisions) [1]. According to a study based on patients at a regional trauma unit in Canada, although the type of impact (lateral vs. non-lateral) had no association with mortality, patients suffered more severe chest and abdominal injuries in lateral collisions than in non-lateral collisions [2]. A comparison of the prevalence of different types of chest and abdominal injuries between different types of collision revealed that hemothorax, lung contusion, splenic injury, kidney injury, retroperitoneal injury, and pelvic fracture were more common in lateral collisions than in non-lateral collisions [2]. In considering passenger injuries and fatalities, the struck side of the vehicle (i.e., either the near side or far side) is a remarkable factor. A study of crash environments suggested that, in lateral collisions, the likelihood of being struck on one side of the vehicle is nearly the same as that of being struck on the other side [3]. However, in lateral collisions, fatalities are more often observed on the near side than on the far side. It is reported that passengers wearing a three-point seatbelt who died in a lateral motor vehicle collision, had severe injuries with an Abbreviated Injury Scale (AIS) score of 3 or more at a rate three times higher for near-side collisions than for far-side collisions [4]. Delta-V is a measure of the near-instantaneous change in velocity sustained by a vehicle during a collision. It was calculated by trained crash researchers using measurable factors, such as vehicle stiffness in an exterior crash. When comparing the prevalence of injury based on National Automotive Sampling System/Crashworthiness Data System (NASS/CDS) data, the injury rates across all delta-V values for lateral collisions are higher for near-side impacts than for far-side impacts. Regarding injury severity, the injury rates for near-side impacts are reported to be between 2.2 and 2.4 times than those for far-side impacts [3]. Occupants involved in far-side lateral collisions were reported to be less likely to be seriously injured because of the less direct impact from vehicle intrusion [5,6]. For vehicle passengers involved in a lateral collision with a delta-V greater than 48 km/h, it was found that the prevalence of suffering from injuries scoring higher than 3 on the Maximum Abbreviated Injury Scale was 84% among near-side occupants and 48% among far-side occupants [7].

A side airbag is a safety system that reduces the severity of injury to the occupant. Although the risk faced by passengers involved in near-side lateral collisions is understood, and despite the presence of side airbags for injury prevention, passengers involved in far-side lateral collisions also suffer serious and fatal injuries. A few reports have suggested the characteristics of passenger injuries in far-side collisions [3,8,9,10]. As shown in Table 1, common injuries were determined in a collision test using a dummy or postmortem human subjects; injury risk curves or injury rates for different levels of severity according to collision velocity are presented using NASS-CDS data. Hostetler et al. presented injury risk curves for AIS scores of 2 to 5 or more in far-side collisions using the NASS/CDS dataset [8]. However, no studies have determined the factors affecting the fatality of passengers involved in far-side lateral collisions. 

The objective of this study was to determine the independent predictive factors of the fatality of motor vehicle passengers involved in far-side lateral collisions.

## 2. Materials and Methods

### 2.1. Study Design and Patient Selection

To achieve the objective of the present study, we considered that analyses of passengers involved in far-side lateral collisions in the real world are indispensable. For greater reliability and accuracy, both injury data with a medical perspective and collision data with mechanical engineering and environmental perspectives are required. Therefore, we focused on a large national motor-vehicle collision database. 

This observational study was a retrospective analysis of the dataset of the National Automotive Sampling System/Crashworthiness Data System (NASS/CDS). The NASS/CDS dataset is generated by the United States National Highway Traffic Safety Administration. It is a publicity available, de-identified dataset that provides data for approximately 5000 collisions every year. The database includes collisions in which at least one of the cars, light trucks, vans, or sports utility vehicles involved was damaged and needed to be towed from the scene. The data in each case were collected from interviews with the people involved, police records, medical records, vehicle inspection, crash-site inspection, and photographs. The raw data can be downloaded via the FTP site of the NASS/CDS. Owing to the anonymous and retrospective nature of this study, which used a database open to the public, the requirements for informed consent and approval by an institutional ethics committee were waived.

Collisions involving 10,215 individuals were registered in the NASS/CDS dataset between 2002 and 2015. In this study, we included individuals who were seated in the front-left or rear-left (second row) seat and were impacted from the right (in a far-side collision). A far-side collision was defined as a vehicle collision with any object where the principal direction of force was between two o’clock and four o’clock (45–135 degrees). Previously, for rear-seated passengers, trends similar to those for front-seat passengers were observed, in that near-side seated passengers have a higher mortality rate and a higher severity of injuries than far-side seated passengers [11,12]. We therefore included the passengers seated in the second row.

We excluded individuals with injuries having an AIS score less than 2 and a height less than 140 cm. We created two datasets for the current study comprising fatal and non-fatal events.

### 2.2. Data Selection

The following information was collected from the database for each person involved in a collision:(1)General characteristics including age, sex, height, and weight. The body mass index (BMI) was also calculated as weight [kg]/height^2^ [m];(2)Seatbelt use;(3)Airbag deployment (front airbag and lateral airbag);(4)Total changes in vehicle velocity (delta-V total, DVTOTAL). After delta-V values in lateral and longitudinal directions were obtained, DVTOTAL was calculated as the square root of the sum of squares of these;(5)Degree of intrusion in the collision;(6)Level of consciousness as described using the Glasgow Coma Scale (GCS). The GCS evaluates a person’s eye opening (E), verbal response (V), and motor response (M). E + V + M is calculated and expressed on a scale from 3 (coma) to 15 (clear);(7)Severity of occupant injury as described using the AIS score. The AIS score is used to categorize the injury type and anatomical severity in each body region on a scale from 1 (minor) to 6 (clinically untreatable);(8)Injury Severity Score (ISS): The ISS is useful for assessing the severity of multiple injuries. The ISS is the sum of the squares of the highest AIS score in each of the three most severely injured body regions. The maximum value is 75, and a higher number indicates greater severity.

### 2.3. Statistical Analysis

The data were summarized as proportions or frequencies for categorical variables. To summarize continuous variables, we used the median and interquartile range for values with a non-normal distribution. Chi-square tests were conducted to compare prevalence between groups. To find the differences in values between groups, a Mann–Whitney test was conducted for values with a non-normal distribution. A *p* value of <0.05 was considered statistically significant. We performed multivariable logistic regression analysis with the forced input method to identify which variables were independently associated with fatality. We conducted a Hosmer–Lemeshow test to determine the goodness of fit of the regression models. In that test, a higher probability indicates better fit. Additionally, we calculated pseudo R^2^ (Nagelkerke R^2^) as an index of the proportion explainable by the regression equation; a larger R^2^ indicates a better model.

The analyses were performed with IBM SPSS version 23 (IBM Corp., Armonk, NY, USA).

## 3. Results

### 3.1. General Characteristics

For a period of 14 years, 325 non-fatal and 86 fatal passengers with an AIS score of 2 or more were collected for analysis. The median age of the passengers was 34 years, and passengers in their twenties accounted for 28% of the total (Figure 1). The general characteristics and background of the collisions are summarized in Table 1. Of the 411 passengers, 55% were male. The passengers had a median height of 170 cm, a median weight of 77 kg, and a median BMI of 25.8. The event sequence number, that is the number of collisions, was 1, and the median model year of the car was 2000. The seatbelt-use rate was 50.5%, the median DVTOTAL was 34 km/h, and the median degree of intrusion was 228 cm. The frontal airbag deployment rate was 29.8% and the lateral airbag deployment rate was 8.5%. Regarding the status of passengers, the median GCS was 14, and for median AIS scores were 2 for the head and zero for other body regions. The median ISS was 11.

### 3.2. Comparison between Fatal and Non-Fatal Groups

Table 2 shows that there was a significantly higher proportion of men in the fatal group than in the non-fatal group, age was significantly higher in the fatal group than in the non-fatal group, and weight and BMI were significantly higher in the fatal group than in the non-fatal group. Regarding the collision details, DVTOTAL and the degree of intrusion were significantly higher in the fatal group than in the non-fatal group. Regarding injury severity, AIS scores were significantly higher in the fatal group than in the non-fatal group for the head, chest, abdomen, spine, and lower extremities. ISS was significantly higher in the fatal group than in the non-fatal group.

### 3.3. Factors Affecting Fatality

To identify variables that were independently associated with fatality, we performed multivariable logistic regression analyses with the predictive variables of sex, age, BMI, seatbelt use, DVTOTAL, degree of intrusion, lateral airbag deployment, and AIS score (for the head, neck, chest, abdomen, and lower extremity). We considered that these predictive factors may affect fatalities on the basis of previous studies. The results showed that female (Ref, male) (odds ratio [OR], 0.396), age (OR, 1.029), BMI (OR, 1.057), DVTOTAL (OR, 1.031), head AIS score (OR, 1.679), chest AIS score (OR, 1.330), and abdomen AIS score (OR, 1.294) were independent predictors of fatality (Table 3). The Hosmer–Lemeshow test indicated a good fit (*p* = 0.917), and the Nagelkerke R^2^ value was 0.466.

## 4. Discussion

Vehicles sold in the US are designed to meet Federal Motor Vehicle Safety Standards which address issues related to lateral collision protection. Vehicle manufacturers are required to evaluate the applied forces to the dummy model sitting on the vehicle seat with dynamic tests simulating a car being struck in the side by another passenger car and the side of the vehicle struck to a pole. Lateral collision regulation had focused on protecting the near-side occupants because occupants in the struck side suffered from more severe injuries than those on the far side. Therefore, few countermeasures are available that specifically aim to improve far-side occupant protection. The New Car Assessment Program in Europe (Euro-NCAP) created a five-star rating system to evaluate vehicle safety based on a series of vehicle tests [13]. This is a comprehensive and objective evaluation that exceeds the minimum legal requirements. A high number of stars indicates not only that the test result is good but also that safety equipment on the tested model is applicable for all vehicle users in Europe. In 2020, the importance of far-side lateral collisions was acknowledged, and far-side collision testing and evaluation protocols were included [14]. In the evaluation, additional tests are performed on a complete vehicle body attached to a sled performing the accelerations experienced by the dynamic vehicle tests. The extent to which the dummy seated on the far side moves towards the impacted side of the vehicle is measured. At the same time, forces and decelerations in the head, neck chest and abdomen of the dummy are measured. Based on these results, the score is calculated from full points, for very limited excursions and lower values of forces and decelerations to no points, where the excursion is extreme and there are higher forces and decelerations. Therefore, car manufacturers are now addressing the safety of passengers involved in far-side lateral collisions and are considering countermeasures to mitigate injuries in such collisions. We believe the present results will be useful for engineers and vehicle manufacturers in addition to forensic scientists analyzing motor vehicle collision (MVC) injuries. Although there have been a few reports suggesting common injuries using crash dummies or postmortem human subjects, as well as risk curves or injury rates according to collision velocity using NASS/CDS data, the factors influencing fatalities among passengers involved in far-side lateral collisions have been unclear. This was the first study to determine predictive factors of fatality in such collisions using detailed information about collision characteristics and injuries.

Among passengers in both the fatal and non-fatal groups, injuries were most severe for the head followed by the chest. The median AIS scores of the head and chest were significantly higher among fatal passengers, being 4 and 3, respectively. In a study on the risk curves for each AIS score, an increase in lateral delta-V resulted in an increased risk of injury [3,8]. In the present study, the median delta-V for fatal passengers was 52.5 km/h, significantly higher than that for non-fatal passengers, 31.0 km/h. A previous study suggested that the delta-V values corresponding to a 50% and 85% risk of suffering injuries having a maximum AIS score of 3 or more were less than 30 and 50 km/h, respectively [9]. Our result is in good agreement with this previous result. As DVTOTAL was selected as a predictive factor for fatality, collision velocity was a good predictor, even for a far-side lateral collision. Our results suggest that a further reduction of the collision velocity through the implementation of pre-crash safety equipment would reduce fatalities in far-side collisions.

Our results suggest that the AIS score for the head was the most prominent predictive factor, with an OR of 1.679, followed by that of the chest (OR, 1.330) and that of the abdomen (1.294). It is well known that the head and chest are critical body regions of passengers in far-side lateral collisions [15]. Reconstructions of far-side lateral collisions with post-mortem human subjects (PMHSs) at a sled acceleration of 60 km/h revealed injuries having an AIS score of 2 or 3 in the neck for five out of six PMHSs, in the chest for five out of six PMHSs, and in the abdomen for two out of six PMHSs [10]. Detailed injuries were ruptures of the disc and dislocations of the cervical spine, rib fractures, spleen injuries, and liver injuries. The mechanisms of injury were related to the kinematics of the passengers at collision. The kinematics depended on the magnitude and direction of the applied force and subsequent interaction of the occupant with the interior structure of the vehicle. To prevent fatalities in far-side lateral collisions, specific effort is needed to avoid the occurrence of these injuries by improving the safety of the interior of the vehicle. Additional seatbelt systems or a side airbag that deploys between seats might be considered for practical use. Recently, finite element models of the human body were developed to investigate injuries and the crashworthiness of motor vehicles at a level of detail difficult to reach with an anthropomorphic test device. Injury in a near-side impact crash was predicted through simulation using a model of the human body [16], and further studies on the mechanisms of head, chest, and abdominal injuries, and proposed safety equipment, are thus required for far-side lateral collisions. 

Although OR is not as high, age was a significant predictor of fatality with the smallest *p* value. This result is in good accordance with the previous results regarding injury risk curves in far-side lateral motor vehicle collisions showing that age increased risks of having injuries of AIS 2 through 5 or more [8]. The previous research also suggested that age increased the risk for thoracic, spinal, or pelvic injuries [8]. These results might be owing to age-related physical changes, such as muscle weakness, osteoporosis, and reduction in the organ system. Furthermore, a systematic review and meta-analysis of trauma patients reported that older patients had higher mortality rates than younger patients [17,18]. Older patients have higher mortality and morbidity than younger patients, even in MVCs of equal severity [19,20]. Therefore, our study confirmed that even in far-side lateral collisions, aged passengers have a higher risk of fatalities.

In this study, seatbelt use was not a predictive factor of mortality. A previous study revealed that the use of a lap and shoulder belt reduced the risk of injuries between an AIS 2 and 5 or more in far-side collisions [8]. Another study based on the National Automotive Sampling System General Estimate System suggested that seatbelts were highly protective against serious or fatal injuries in lateral collisions after controlling for the seating position and crash-impact type [11]. However, in another study on far-side collisions, among restrained passengers, 72% of abdominal injuries were due to compression exerted by a lap belt [9]. Although this was not a significant influencing factor for fatality in far-side lateral collisions, seatbelt use was related to the kinematics of passengers in lateral collisions. 

The degree of intrusion of the passenger compartment has been used as the best index for the injury severity of the passenger [6,21,22,23]. In a near-side lateral collision, because there is less vehicle structure between the striking force and the occupant, there is appreciable passenger compartment intrusion and subsequent direct impact onto the occupant’s chest and abdomen. However, in this study, the degree of intrusion was not a significant predictive factor of injury severity. Because the passenger on the far side was less directly impacted, the degree of intrusion was not an index for predicting fatality. 

A typical torso airbag is a type of side airbag installed in a door that deploys to absorb the energy being transferred between the vehicle structure and the occupant. Curtain airbags deploy from the roof rail to prevent ejection of the passenger and absorb energy being transferred to the occupant’s head. Previous studies have reported that side airbags reduce the risk of death by approximately 30–40% in lateral collisions and reduce the risk of head injury having an AIS score of 2 or more by 30% [24,25]. However, side airbags were not selected as a significant predictive factor of fatality among far-side passengers in the present study. The result is in good agreement with the result of a previous study showing that curtain and side airbags did not provide benefits in far-side impacts [26].

This study has several limitations. First, the presence of same-row occupants was not examined, yet there are interactions between adjacent occupants in a lateral collision. However, a previous study based on NASS/CDS data suggested that the presence of a same-row occupant was not a significant factor affecting injury risk curves [8]. To confirm the present results, further studies including the factor of same-row occupants need to be conducted. Second, the average age of an injured occupant in the present study was 34 years. In developed countries, populations tend to be aging, and fatalities in motor-vehicle collisions depend on age. For each body region, material properties change according to age. Therefore, the present results would not apply to children or aged passengers. Indeed, a study based on real-world near-side lateral collisions suggested the risk of chest injuries having an AIS score of 3 or more for aged passengers (60 years or older) was approximately four times that for passengers aged 10 to 59 [27]. In addition, our study suggested that age was a predictive factor for fatality although the OR was as low as 1.029. Further study on aged passengers would allow a comparison of results with our results. Third, because the database used in the present study comprised vehicle collisions in the US, it did not reflect the traffic environment and characteristics of other countries;(e.g., different countries have different body sizes, vehicle sizes and speed limits). Future research using other databases should be conducted to provide comparative results. Given the limitations of this study, further studies are required in both developed and developing countries, where lateral collision is the leading type of fatal collision. The use of accumulated data of injuries with different ages and body sizes and under different collision circumstances (e.g., vehicle type, collision velocity) might serve to break through the limitations of our current study.

## 5. Conclusions

Using a national crash database, we examined 411 passengers involved in far-side lateral collisions. When comparing fatal and non-fatal passengers, DVTOTAL and the degree of intrusion were significantly higher in the fatal group than in the non-fatal group. Regarding injury severity, AIS scores were significantly higher in the fatal group than in the non-fatal group for the head, chest, abdomen, spine, and lower extremities. ISS values were significantly higher in the fatal group than those in the non-fatal group. 

In multivariable logistic regression analyses, female sex (Ref, male) (OR, 0.396), age (OR, 1.029), BMI (OR, 1.057), DVTOTAL (OR, 1.031), head AIS score (OR, 1.679), chest AIS score (OR, 1.330), and abdomen AIS score (OR, 1.294) were independent predictors of fatality. This is the first report to determine factors affecting fatality among passengers involved in far-side lateral collisions. Improving the safety of the vehicle interior, such as by including additional seatbelt systems, or a side airbag that deploys between seats, might help to avoid fatalities, and reduce injury severity. 

## Figures and Tables

**Figure 1 healthcare-11-01496-f001:**
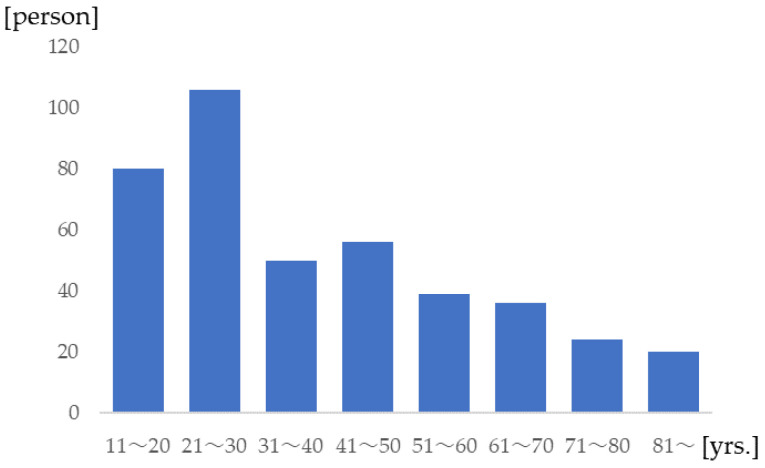
Age distribution of the passengers.

**Table 1 healthcare-11-01496-t001:** Previous studies reporting passengerss injuries in far-side lateral collisions.

Reference No.	Year	Summary	Materials
[3]	2015	Injury rates of maximum AIS of 3 or more (3+), AIS4+ or more were determined according to vehicle types or collision velocity.	NASS/CDS data
[8]	2020	Injury risk curves for AIS2+, 3+, 4+, 5+ were determined and affective factors for risks were determined.	NASS/CDS data
[9]	2001	Injury risk curves for maximum AIS3+ were determined and effect of seatbelt was examined.	NASS/CDS data
[10]	2019	Common injuries were determined by the reappearance of the lateral collision with a crash dummy and postmortem human subjects.	Crash dummy and postmortem human subjects

**Table 2 healthcare-11-01496-t002:** Comparison of fatal and non-fatal groups.

	All (N = 411)	Non-Fatal (N = 325)	Fatal (N = 86)	*p*
Sex (M:F)	224:187	168:157	56:30	0.026
Age (yrs.)	34 (23, 54)	32 (22, 54)	42 (25, 56)	0.045
Height (cm)	170 (163, 178)	170 (163, 178)	173 (168, 180)	0.055
Weight (kg)	77 (65, 91)	77 (64, 90)	82 (68, 97)	0.0036
BMI	25.8 (22.5, 30.4)	25.5 (22.2, 29.7)	27.0 (23.1, 33.7)	0.023
Seatbelt use	50.50%	51.4%	46.5%	0.48
DVTOTAL (km/h)	34 (23, 50)	31 (22, 45)	52.5 (35, 66)	<0.001
Degree of intrusion (cm)	228 (164, 279)	223 (160, 275.5)	241 (190, 308)	0.014
Lateral airbag deployment	8.50%	8.3%	9.3%	0.77
GCS	14 (2, 15)	15 (6, 15)	1.5 (1, 3)	<0.001
Head AIS	2 (0, 3)	2 (0, 2)	4 (2, 5)	<0.001
Face AIS	0 (0, 0)	0 (0, 0)	0 (0, 0)	0.16
Neck AIS	0 (0, 0)	0 (0, 0)	0 (0, 0)	NaN
Chest AIS	0 (0, 3)	0 (0, 3)	3 (0, 4)	<0.001
Abdomen AIS	0 (0, 0)	0 (0, 0)	0 (0, 2)	<0.001
Spine AIS	0 (0, 2)	0 (0, 0)	0 (0, 2)	<0.001
Upper extremity AIS	0 (0, 0)	0 (0, 0)	0 (0, 2)	0.57
Lower extremity AIS	0 (0, 0)	0 (0, 0)	0 (0, 2)	0.0088
ISS	11 (5, 26)	9 (5, 18)	35 (19, 57)	<0.001

Summarized data are shown as medians and interquartile ranges.

**Table 3 healthcare-11-01496-t003:** Results of multivariable logistic regression analyses.

Variable	Odds Ratio	95% Confidence Interval	*p* Value
Sex	0.396	0.200―0.783	0.008
Age	1.029	1.012―1.046	0.001
BMI	1.057	1.008―1.109	0.023
Seatbelt use	0.936	0.476―1.839	0.85
DVTOTAL	1.031	1.012―1.051	0.002
Degree of intrusion	1.001	0.997―1.006	0.66
Lateral airbag deployment	0.622	0.189―2.043	0.43
Head AIS	1.679	1.400―2.013	0
Chest AIS	1.33	1.090―1.624	0.005
Abdomen AIS	1.294	1.003―1.668	0.047
Lower Extremity AIS	1.117	0.829―1.506	0.47

## Data Availability

The data presented in this study are available upon request from the corresponding author.

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
