# Peer review of "Predictive Factors of the Fatality of Motor Vehicle Passengers Involved in Far-Side Lateral Collisions: A National Crash Database Study"

_healthcare, 2023, doi:10.3390/healthcare11101496_

Round 1

Reviewer 1 Report

Dear Authors,

Please take into consideration my following concerns:

In my opinion, the manuscript is within the scope of the journal. The structure is correct and consists of all of the necessary sections. However, the novelty and approach are not satisfactory from my perspective. The novelty of the work must be clearly addressed in the Abstract and discussed widely in the Discussion section with a comparison to different authors’ results and methods. Unfortunately, you described the results but the approach is not satisfactory to be published in such a prestigious journal as Healthcare. 

Basically, I am concerned about the methodological part of the paper. The theoretical assumptions are obscure and very general. Please describe in more detail how you achieved your goals.

Moreover, you need to more focus on recommendations and future research arises from the limitations of your study and results gained during the experiment. For that moment it is only one sentence at the end of section 4.

In my opinion, you should also include numerical results in the conclusion section to support the main findings. The presented manuscript does not contain them in the mentioned part of the manuscript.

All references are provided applicable, however, I suggest extracting these key references (with other authors' approaches) from the text and putting them into a table focusing on weaknesses. Otherwise, it is very difficult to follow the Introduction section.

Please consider rebuilding Tables 1-3 to keep the same fonts and conventions in the paper.

To sum up, please analyze the above concerns. I recommend rebuilding the manuscript and taking into consideration my comments.

Regards,

I did not notice any serious language and grammatical errors.

Author Response

Responses to the comments of Reviewer 1

Thank you for your careful review of our manuscript. Your comments have enabled us to improve the quality of our manuscript.

In my opinion, the manuscript is within the scope of the journal. The structure is correct and consists of all of the necessary sections. However, the novelty and approach are not satisfactory from my perspective. The novelty of the work must be clearly addressed in the Abstract and discussed widely in the Discussion section with a comparison to different authors’ results and methods. Unfortunately, you described the results but the approach is not satisfactory to be published in such a prestigious journal as Healthcare.

Following your suggestion, we have added a sentence in Abstract, as follows (Lines 27—29): “This is the first report to determine factors affecting fatality in passengers involved in far-side lateral collisions.”

We have also added a following description in the last of first paragraph of the Discussion (Lines 241—246): “Although there have been a few reports suggesting common injuries using crash dummies or postmortem human subjects, as well as risk curves or injury rates according to collision velocity using NASS/CDS data, the factors influencing fatalities among passengers involved in far-side lateral collisions have unclear. This was the first study to determine predictive factors of fatality in such collisions using detailed information about collision characteristics and injuries.” 

Basically, I am concerned about the methodological part of the paper. The theoretical assumptions are obscure and very general. Please describe in more detail how you achieved your goals.

In accordance with your suggestion, we have added a description at the beginning of the Material and Methods section (Lines 92—96): “To achieve the objective of the present study, we considered that analyses of passengers involved in far-side lateral collisions in the real-world are indispensable. For greater reliability and accuracy, both  injury data with a medical perspective and collision data with mechanical engineering and environmental perspectives are required. Therefore, we focused on a large national motor vehicle collision database.”

Moreover, you need to more focus on recommendations and future research arises from the limitations of your study and results gained during the experiment. For that moment it is only one sentence at the end of section 4.

In the original version, we provided recommendations and needs for future research in the limitations (Line 3201—323, 330—331, and 334—335). However, in accordance with your suggestion, we have added the following sentences in the Discussion section: “Given the limitations of this study, further studies are required in both developed and developing countries, where lateral collision is the leading type of fatal collision. Use of accumulated data of injuries with different ages and body sizes and under different collision circumstances (e.g., vehicle type, collision velocity) might serve to break through the limitations of our current study”. 

In my opinion, you should also include numerical results in the conclusion section to support the main findings. The presented manuscript does not contain them in the mentioned part of the manuscript.

We have rewritten the Conclusion, as follows: “Using a national crash database, we examined 411 passengers involved in far-side lateral collisions. When comparing fatal and non-fatal passengers, DVTOTAL and the degree of intrusion were significantly higher in the fatal group than in the non-fatal group. Regarding injury severity, AIS scores were significantly higher in the fatal group than in the non-fatal group for the head, chest, abdomen, spine, and lower extremities. ISS values were significantly higher in the fatal group than those in the non-fatal group.

In multivariable logistic regression analyses, female sex (Ref, male) (OR 0.396), age (OR 1.029), BMI (OR 1.057), DVTOTAL (OR 1.031), head AIS score (OR 1.679), chest AIS score (OR 1.330), and abdomen AIS score (OR 1.294) were independent predictors of fatality. This is the first report of factors affecting fatality among passengers involved in far-side lateral collisions. Improving the safety of the vehicle interior, such as by including additional seatbelt systems or a side airbag that deploys between seats, might avoid fatalities and reduce injury severity.” (Lines 342—354)

All references are provided applicable, however, I suggest extracting these key references (with other authors’ approaches) from the text and putting them into a table focusing on weaknesses. Otherwise, it is very difficult to follow the Introduction section.

Following your suggestion, we have added a table that shows the approaches of other authors and we have corrected the descriptions in the Introduction section (Lines 80—83): “As shown in Table 1, common injuries were determined in a collision test using a dummy or postmortem human subjects; injury risk curves or injury rates for different levels of severity according to collision velocity are presented using NASS-CDS data.”

Please consider rebuilding Tables 1-3 to keep the same fonts and conventions in the paper.

We have added an additional table, and renumbered the tables. We have recreated Tables 2—3 while maintaining the formatting requirements and conventions in the paper.

To sum up, please analyze the above concerns. I recommend rebuilding the manuscript and taking into consideration my comments.

Reviewer 2 Report

The paper is very well written, and contributes predictive factors using multivariable logistic regression analysis scheme for the risks faced by passengers in far-side lateral collision, which enables intuitively see the influence of different factors. However, there are some problems, which must be solved before it is considered for publication. If the following problems are well-addressed, the essential contribution of this paper will be important for far-side lateral collisions.

In the Abstract section, the overall structure and logic are very good. But in some places the reference is unclear. For example, the conclusion in line 24 to 27 is the result of logistic regression after reading the later text, but the meaning of these data cannot be accurately understood at the first reading. Therefore, it is suggested to integrate with the sentence in line 23 into one sentence.

In the Introduction section, the author introduces the problem clearly, cites enough previous work, and gives an overview of the current development level of the problem. However, please make a clear definition of ‘restrained passengers’ in line 48-49. I'm not sure if you mean ‘restrained passengers wearing seat belts’. Also, I would like to know by what criteria ‘between 2.2 to 2.4 times’ in line 55 is measured. As we know from the above, it must not be AIS score in 3 points , then what is it?

In the section of Materials and Methods, the paragraphs are reasonably organized and the language description is clear. I personally suggest that descriptions of indicators such as MBI, DVTOTAL, GCS, AIS and ISS should be added. In addition, in line 90, ‘between 2 o 'clock and 4 o 'clock (45-135 degrees)’, actually, it is 60-120 degrees. Please describe 45-135 degrees in more appropriate language.

In the Result section, I have a few suggestions:

1. Whether Table 1 and Table 2 can be integrated into one table. Adding a column before the ‘Non-fatal’ in Table 2 could express the contents of ‘All’ and make table more intuitive.

2. In the description of lines 142-148, we know that some data are averages and some data are medians. For the sake of prudence, please indicate whether they are average or median when they appear in the later tables.

3. I wonder if the description of lines 160 to 167 is directly observed from Table 2. Among these, there are lots of ‘significantly higher’. Please explain how to define this. Because the MBI and ‘the degree of intrusion" in Table 2 are not significantly higher.

4. For the prediction variables in line 175-177, it is hoped that they can be arranged in tabular order, which is easier for readers to read and compare. Moreover, ‘forced input’ is not reflected on the form, I suggest it could be expanded the explanation.

There are the most language descriptions in Discussion and Conclusion section of the whole article, and it is recommended that the current manuscript needs to be polished by a native English speakers or professional language editing services. The author's conclusions on the influence of ‘seat belt use’, ‘the degree of passenger compartment instrusion’ and ‘side airbag’ are vague, which is unfavorable to the final Conclusion. Please give a clearer conclusion.

Here are some suggestions for details in these two part. Please specify what is the meaning of ‘readily available to all consumers in Europe’ in line 199-200, and what evaluation criteria are included in line 201 ‘far side collision test and evaluation protocol are included’. In addition, please revise the description of the score in lines 206 to 208. And please add the full name of ‘MVC injuries’ in line 212, I think it is ‘Motor Vehicle Collision’. There are multiple ‘out of six PMHSs’ in line 231 and 232, which is suggested to modify the expression. The fractional interval of AIS is not clearly stated in line 249.

Author Response

Responses to the comments of Reviewer 2

Thank you for your careful review of our manuscript. Your comments have enabled us to improve the quality of our manuscript.

The paper is very well written, and contributes predictive factors using multivariable logistic regression analysis scheme for the risks faced by passengers in far-side lateral collision, which enables intuitively see the influence of different factors. However, there are some problems, which must be solved before it is considered for publication. If the following problems are well-addressed, the essential contribution of this paper will be important for far-side lateral collisions.

In the Abstract section, the overall structure and logic are very good. But in some places the reference is unclear. For example, the conclusion in line 24 to 27 is the result of logistic regression after reading the later text, but the meaning of these data cannot be accurately understood at the first reading. Therefore, it is suggested to integrate with the sentence in line 23 into one sentence.

In accordance with your suggestion, we have integrated lines 23 and 24 to 27 as follows: “In multivariable logistic regression analysis, variables independently associated with fatalities were: female sex (Ref, male) (odds ratio [OR] 0.396), age (OR 1.029), body mass index (OR 1.057), total delta-V (OR 1.031), head AIS score (OR 1.679), chest AIS score (OR 1.330), and abdomen AIS score (OR 1.294) .

In the Introduction section, the author introduces the problem clearly, cites enough previous work, and gives an overview of the current development level of the problem. However, please make a clear definition of ‘restrained passengers’ in line 48-49. I'm not sure if you mean ‘restrained passengers wearing seat belts’. Also, I would like to know by what criteria ‘between 2.2 to 2.4 times’ in line 55 is measured. As we know from the above, it must not be AIS score in 3 points , then what is it?

Following your suggestion, we have corrected the sentence as follows (Lines 52—55): “It is reported that passengers wearing a three-point seatbelt who died in a lateral motor vehicle collision, had severe injuries with an Abbreviated Injury Scale (AIS) score of 3 or more at a rate three times higher for near-side collisions than for far-side collisions.”

 In accordance with your suggestion, we have corrected the sentence as follows (Lines 61—62): “Regarding injury severity, the injury rates for near-side impacts are reported to be between 2.2 and 2.4 times those for far-side impacts [3].”

In the section of Materials and Methods, the paragraphs are reasonably organized and the language description is clear. I personally suggest that descriptions of indicators such as MBI, DVTOTAL, GCS, AIS and ISS should be added. In addition, in line 90, ‘between 2 o 'clock and 4 o 'clock (45-135 degrees)’, actually, it is 60-120 degrees. Please describe 45-135 degrees in more appropriate language.

  Following your suggestion, we have corrected the sentences regarding BMI as follows (Line126): “Body mass index (BMI) was also calculated as: weight [kg]/height2 [m].” and DVTOTAL (Lines 125—129): “Delta-V is the measure of the near-instantaneous change in velocity sustained by the vehicle during the collision. It has been calculated by trained crash researchers using measurable factors such as vehicle stiffness in an exterior crash. (Lines 55—58): After delta-V values in lateral and longitudinal directions were obtained, DVTOTAL was calculated as the square root of the sum of squares of these.” (Lines 129—131) Also following your suggestion, we have corrected sentences regarding the ISS as follows (Lines 141—143): “The ISS is useful for assessing the severity of multiple injuries. The ISS is the sum of the squares of the highest AIS score in each of the three most severely injured body regions.

  For the GCS and AIS, descriptions of these indicators were already provided in the original manuscript (Lines 135—140).

  We apologize for our error regarding the degrees. We have corrected this to 60—120 degrees.

In the Result section, I have a few suggestions:

  1. Whether Table 1 and Table 2 can be integrated into one table. Adding a column before the ‘Non-fatal’ in Table 2 could express the contents of ‘All’ and make table more intuitive.

According to your suggestion, we have combined Table 1 and 2 in the original manuscript into one table (Table 2).

  1. In the description of lines 142-148, we know that some data are averages and some data are medians. For the sake of prudence, please indicate whether they are average or median when they appear in the later tables.

In this article, all summarized data are shown as median and interquartile range. We have indicated that the summarized data are shown as median and interquartile range below Table 2.

  1. I wonder if the description of lines 160 to 167 is directly observed from Table 2. Among these, there are lots of ‘significantly higher’. Please explain how to define this. Because the MBI and ‘the degree of intrusion" in Table 2 are not significantly higher.

We discussed this in the Statistical analysis as follows: “A p value of <0.05 was considered statistically significant.” (Line 153—154) For the above reason we consider that the BMI and degree of intrusion in Table 2 were significantly higher in the fatal group.

  1. For the prediction variables in line 175-177, it is hoped that they can be arranged in tabular order, which is easier for readers to read and compare. Moreover, ‘forced input’ is not reflected on the form, I suggest it could be expanded the explanation.

In accordance with your suggestion, we have corrected the sentence as follows (Lines 196—199): “To identify variables that were independently associated with fatality, we performed multivariable logistic regression analyses with the predictive variables of sex, age, BMI, seatbelt use, DVTOTAL, degree of intrusion, lateral airbag deployment, and AIS score (for the head, neck, chest, abdomen, and lower extremity).”

“Forced input” means the forced input method in multivariable logistic regression analysis. We corrected the sentence in the Statistical analysis section, as follows: “We performed multivariable logistic regression analysis with the forced input method to identify which variables were independently associated with fatality.” (Lines 154—156)

There are the most language descriptions in Discussion and Conclusion section of the whole article, and it is recommended that the current manuscript needs to be polished by a native English speakers or professional language editing services. The author's conclusions on the influence of ‘seat belt use’, ‘the degree of passenger compartment instrusion’ and ‘side airbag’ are vague, which is unfavorable to the final Conclusion. Please give a clearer conclusion.

The original version of our manuscript was edited by English language experts from an editing service. However, in accordance with your suggestion, the revisions to the manuscript haves undergone further English editing.

The description in the conclusion has been corrected according to Reviewer 1.

Here are some suggestions for details in these two part. Please specify what is the meaning of ‘readily available to all consumers in Europe’ in line 199-200, and what evaluation criteria are included in line 201 ‘far side collision test and evaluation protocol are included’. In addition, please revise the description of the score in lines 206 to 208. And please add the full name of ‘MVC injuries’ in line 212, I think it is ‘Motor Vehicle Collision’. There are multiple ‘out of six PMHSs’ in line 231 and 232, which is suggested to modify the expression. The fractional interval of AIS is not clearly stated in line 249.

 Following your suggestion, we have corrected the sentence as follows (Lines 226—228): “A high number of stars indicates that not only that the test result is good but also that safety equipment on the tested model is applicable for all vehicle users in Europe.” Evaluation criteria comprise the degree of movement of the dummy as well as forces and decelerations of each part of the dummy in sled tests. This was described in Line 230—237 of the original manuscript. However, following your suggestion, we have added, “In the evaluation,” at the beginning of the sentence in Line 230.The terms “MVC” and “PMHS” have been rewritten. The AIS is scored from 1 to 6. Therefore, AIS score is shown as an integer.

Round 2

Reviewer 1 Report

Dear Authors,

Thank you for your responses. From my perspective, the paper is much better after your adjustments. 

I wish you good luck for your future research.

Regards,

Reviewer 2 Report

Thanks for the efforts of responding my comments. You have addressed my concerns to a satisfactory level.